# Plant Pathogen Invasion Modifies the Eco-Evolutionary Host Plant Interactions of an Endangered Checkerspot Butterfly

**DOI:** 10.3390/insects12030246

**Published:** 2021-03-15

**Authors:** Paul M. Severns, Melinda Guzman-Martinez

**Affiliations:** Department of Plant Pathology, University of Georgia, Athens, GA 30602, USA; Melinda.Guzman@uga.edu

**Keywords:** biological invasions, *Euphydryas*, *Plantago lanceolata*, *Pyrenopeziza*, dermateaceae

## Abstract

**Simple Summary:**

Taylor’s checkerspot butterfly is a critically endangered species of northwestern North America that has become dependent on an exotic food plant, English plantain, which was acquired over a century ago. In the mid-2000s, a non-native plant pathogen from Europe, invaded Taylor’s checkerspot populations causing English plantain leaves to die in the winter when Taylor’s checkerspot larvae are feeding. We characterized butterfly and larval food plant (native and non-native) timing in Oregon and Washington populations and discovered that the invasive plant disease is active for ~60 days when larvae are feeding in January, February and March. Only one native larval foodplant, the annual *Collinsia parviflora*, can provide food for caterpillars during the time the plantain disease is common. However, this plant is rare in Taylor’s checkerspot habitat and may only be suitably timed to Washington checkerspot populations. Other native perennial larval food plants (*Castilleja*
*levisecta* and likely *C. hispida*) do not appear suitably timed to provide resources throughout the entire Taylor’s checkerspot lifecycle in the low-elevation English plantain dependent populations. Understanding and accounting for the plant population disease dynamics is essential for the long-term conservation of Taylor’s checkerspot butterfly.

**Abstract:**

New plant pathogen invasions typified by cryptic disease symptoms or those appearing sporadically in time and patchily in space, might go largely unnoticed and not taken seriously by ecologists. We present evidence that the recent invasion of *Pyrenopeziza plantaginis* (Dermateaceae) into the Pacific Northwest USA, which causes foliar necrosis in the fall and winter on *Plantago lanceolata* (plantain), the primary (non-native) foodplant for six of the eight extant Taylor’s checkerspot butterfly populations (*Euphydryas editha taylori*, endangered species), has altered eco-evolutionary foodplant interactions to a degree that threatens butterfly populations with extinction. Patterns of butterfly, larval food plant, and *P. plantaginis* disease development suggested the ancestral relationship was a two-foodplant system, with perennial *Castilleja* spp. supporting oviposition and pre-diapause larvae, and the annual *Collinsia parviflora* supporting post-diapause larvae. Plantain, in the absence of *P. plantaginis* disease, provided larval food resources throughout all butterfly life stages and may explain plantain’s initial adoption by Taylor’s checkerspot. However, in the presence of severe *P. plantaginis* disease, plantain-dependent butterfly populations experience a six-week period in the winter where post-diapause larvae lack essential plantain resources. Only *C. parviflora*, which is rare and competitively inferior under present habitat conditions, can fulfill the post-diapause larval feeding requirements in the presence of severe *P. plantaginis* disease. However, a germination timing experiment suggested *C. parviflora* to be suitably timed for only Washington Taylor’s checkerspot populations. The recent invasion by *P. plantaginis* appears to have rendered the ancestrally adaptive acquisition of plantain by Taylor’s checkerspot an unreliable, maladaptive foodplant interaction.

## 1. Introduction

Invasions by new plant pathogens that cause disease on ecologically dominant plants such as American Chestnut Blight (*Cryphonectria parasitica*), Dutch Elm Disease (*Ophiostoma ulmi s.l.)*, and Sudden Oak Death (*Phytophthora ramorum*) can abruptly dismantle and perturb entire natural communities, redirecting ecological interactions and ecosystem functions [1,2,3]. While these plant diseases, and a handful of others, are conspicuous and well-known to have broad reaching impacts on natural ecosystems, they can be treated as the rarest and extraordinary instances of plant disease perturbations on naturally occurring, non-agricultural ecosystems [4,5]. Plant pathogen invasions occurring on plants that are neither competitively dominant nor keystone species could also have similarly disruptive ecological effects while displaying symptoms that are cryptic compared with landscape-wide tree and forest death. *Pyrenopeziza plantaginis* (Dermateaceae), an ascomycete that produces small apothecia and causes necrotic lesions on the dormant, fully-formed leaves of the perennial *Plantago lanceolata* (English plantain—hereafter plantain), an exotic plant to North America [6,7], recently invaded the Pacific Northwest, USA from northern Europe in the mid-2000s via an unknown vector [8]. The invasion of a plant pathogen causing disease on an exotic plant might be considered beneficial, as the pathogen could reduce the relative abundance of a weed [9,10]. However, *P. plantaginis* invasion has brought about a new and significant risk for the long-term persistence and conservation of an endangered species, the Taylor’s checkerspot butterfly (*Euphydryas editha taylori*, Nymphalidae) [11]. 

Prior to the invasion of exotic grasses which coincided with the extinction of some Taylor’s checkerspot populations [12], this species was once one of the most common and abundant butterflies observed in the early spring in western Oregon and Washington, USA, appearing to swarm by the thousands [13]. While native insect interactions with exotic plants are often presumed to be negative, Taylor’s checkerspot has relied on plantain as a larval foodplant since its initial description in 1888 (Edwards) from Vancouver Island, British Columbia, Canada [14]. Furthermore, six of the eight extant Taylor’s checkerspot populations are considered to be plantain-dependent today [12,15]. Plantain is widespread and locally abundant throughout many habitat types [6,7] and despite this, Taylor’s checkerspot remains endangered because it requires a narrow range of habitat conditions for persistence and females rarely disperse far from their natal habitat patches [12,15,16,17,18]. Although in North America plantain is considered an exotic weed, it is widely used as a larval foodplant by butterflies in Europe [19] and several North American butterfly species have independently adopted plantain as a larval foodplant [20,21,22,23]. 

*Pyrenopeziza plantaginis* causes necrosis on the basal leaves of plantain in the autumn, shortly after the rains begin and air temperatures drop. Infections are verified through the presence of tan-rimmed apothecia in the center of necrotic regions [8,11] and *P. plantaginis* appears to be the only pathogen on plantain in Oregon and Washington with these traits. Primary infections presumably arise via splash dispersed spores from the previous year’s inoculum which over-summered on previously infected leaf debris [8]. Fall infection, and possibly secondary infection, continues through the winter and early spring, frequently leading to plantain leaf crown necrosis exceeding 50% of the foliar area up to 100% on each infected plant [11]. It is during the winter and early spring months that *P. plantaginis* disease jeopardizes Taylor’s checkerspot populations, not as adult butterflies (when disease is absent) but as post-diapause larvae. Plantain individuals that are not infected with *P. plantaginis* initiate new leaves that are positioned slightly above the basal leaves in the fall and winter, on which Taylor’s checkerspot post-diapause larvae preferentially feed [11], see [24] Figure 1B. However, when *P. plantaginis* causes disease, leaf initiates are reduced to about 20% of non-infected plants and may be delayed by several months until the late winter and early spring [11]. When leaf necrosis approaches or exceeds 50%, post-diapause larvae are highly unlikely to feed on partially symptomatic leaves, including the green portions that appear asymptomatic [11]. Depending on disease severity and relative abundance, *P. plantaginis* can reduce plantain leaf numbers to levels that cannot support a plantain-dependent Taylor’s checkerspot population [11]. Indeed, severe *P. plantaginis* disease levels (>90% plants infected with >65% mean foliar necrosis) have been associated with Taylor’s checkerspot reintroduction failures in Washington and population extinction in Oregon [11].

Although six of the eight extant Taylor’s checkerspot populations in Oregon and Washington are plantain-dependent, there are native foodplant alternatives such as *Castilleja hispida*, *Castilleja levisecta* (hemi-parasitic perennials via haustorial connections with the host plant, Orobanchaceae), and *Collinsia parviflora* (annual, Veronicaceae) that exist in low abundance within plantain-dependent populations [15]. Due to the recent invasion of *P. plantaginis*, these alternative native larval foodplants may be essential for the future of Taylor’s checkerspot, but the potential for asynchronous developmental timing between plantain, alternative native foodplants, and butterfly development is not adequately known for either Oregon (Beazell Memorial Forest, Cardwell Hill) or Washington (Joint Base Lewis-McChord-2 locations, Sequim, Scatter Creek) plantain-dependent Taylor’s checkerspot populations. Characterization of the phenological interactions between *P. plantaginis* disease development and pre- and post-diapause larval foodplant availability is necessary to fully evaluate the impacts of *P. plantaginis* disease for Taylor’s checkerspot conservation.

## 2. Materials and Methods

### 2.1. Taylor’s Checkerspot Georgraphic Distribution, Developmental Timing, and Life History

Adult Taylor’s checkerspot butterflies emerge from pupae in late March through May at low elevation grassland locations in western Oregon and Washington, USA. The earliest flight times are in the southern part of the butterfly’s range (Oregon) and the latest flights occur two months later at higher elevations (>1500 m elevation) in Washington. We focus on the low elevation populations in Oregon and Washington (Beazell Memorial Forest, Cardwell Hill, Joint Base Lewis–McChord, two locations, Sequim, Scatter Creek), because those are plantain-dependent and the higher elevation sites have low plantain abundance [15]. In plantain-dependent Taylor’s checkerspot populations, females oviposit clusters of 30–120 eggs on the underside of plantain leaves positioned within 5 cm of the ground [25] from April through late May. Eggs develop over two to three weeks and then hatch (May-June), giving rise to clusters of 1st instar larvae that feed gregariously (initially on the oviposition plant and later on nearby plants) through three instars (about 1 month), and then enter diapause for the remainder of the summer and fall (Figure 1). Post-diapause larvae become active and begin feeding in the winter, although the earliest dates of larval feeding are generally not reported as most butterfly field work begins in March when larvae are in their last (5th) instar.

There are two oviposition plants used by Taylor’s checkerspot in extant populations, plantain and *Castilleja hispida* [12,15]. *Castilleja hispida* is naturally uncommon in low elevation Taylor’s checkerspot populations in Washington [15], although recent habitat restoration has featured the supplementation of sites with two Washington species *C. hispida* and *C. levisecta* (a federally listed endangered species) [26,27]. These oviposition plants also serve as the primary, if not sole, larval foodplant for pre-diapause Taylor’s checkerspot as this life history aspect mirrors other *Euphydryas editha* conspecifics [25,28,29]. However, when butterflies are in flight, the young spring leaves of plantain are *P. plantaginis* disease-free and remain so until the fall when plantain is largely dormant but maintains photosynthetically active basal leaves [8]. Consequently, *P. plantaginis* disease is unlikely to influence Taylor’s checkerspot oviposition and pre-diapause larval feeding.

Post-diapause Taylor’s checkerspot larvae feed predominately on the young, newly initiated plantain leaves but may also feed on the native annual *Collinsia parviflora* (Washington only) and the young emerging leaves of *Castilleja* spp. (Washington only) [15] if plant and larval development overlap. *Plectritis congesta* (Valerianaceae), suggested by some to be a post-diapause larval foodplant, does not appear acceptable to western Oregon and Washington post-diapause larvae. Palatability trials (five larvae from each Oregon and Washington populations in 2011) all refused to eat *P. congesta* tissue offered for a continuous 48 h despite a 24-h period being offered no food prior to *P. congesta* introduction (Severns unpublished data).

### 2.2. Butterfly Development, Larval Foodplant Availability, and P. plantaginis Disease

Temporal profiles were constructed to visualize potential negative butterfly-plant-disease interactions by overlaying butterfly developmental stage timings, larval foodplant availability, and the proportion of severely diseased plants averaged over a three-year period (2014 to 2016). Data were generated through close monitoring of butterflies and host plants in the Oregon populations (where site access was reliable) and personal communications with biologists working in Washington populations (where access was limited). Adult Taylor’s checkerspot surveys in both states can be acutely limited by access restrictions and inclement weather, such that some sites receive as little as 3 visits in a flight season, yet others will have seven surveys. This data set presents some challenges as it can be sparse depending upon site and year, so we focus on the broad phenological development signals. The year with the most consistent data was 2015 and this served a base template to stitch together observations and data from the aggregate time span to produce a general profile of development timings. The availability of larval foodplants and butterfly relative abundance were expressed as the mean proportion of the total measured populations sizes arranged by Julian date as a way to stardardize time over multiple years (although we refer to monthly times as a convenient way to interpret the broader seasonal patterns). By expressing relative abundance as a proportion, we focus on the general phenological changes over time and downweight the short-term, nuanced responses to weather and between year differences that we do not have enough data to adequately characterize. Furthermore, this approach moderates fluctuations in disease levels and butterfly population counts, rather than attempting to represent how absolute numbers change over time, as butterflies counts, *P. plantaginis* disease, and plant populations can be patchily distributed and prone to substantial between-year fluctuations. 

Because post-diapause larval feeding was predominately known from anecdotal observations made late in butterfly development, two Taylor’s checkerspot sites near Corvallis, OR, USA were visited on any sunny day with temperatures above freezing beginning in the last week of December (years 2008–2017) to determine when post-diapause larval feeding began. Areas with abundant plantain and where pre-diapause larvae were known to occur the previous spring were subjectively searched for actively feeding, post-diapause Taylor’s checkerspot larvae. Once at least two post-diapause larvae were found, searches ceased for the year as Taylor’s checkerspot post-diapause larvae do not appear to commonly re-enter diapause (Severns pers. obs.) unlike other Pacific Northwest *E. editha* conspecifics [30]. Adult butterfly flight times and pre-diapause larval time periods in Oregon populations were estimated from adult butterfly monitoring reports and through verbal communications with agency biologists, land managers and surveyors. Because adult Taylor’s checkerspot butterfly populations are monitored with different sampling techniques (e.g., distance sampling, grids, transects counts) depending on the state and site, we used the number of butterflies reported at each sampling date (from a combination of surveyor personal communications and year end status reports) and expressed relative abundance at that date as the proportion of the total butterfly numbers counted/estimated that year. 

Permanent monitoring plots (fifteen 2 × 2 m quadrats) were established at three sites near Corvallis, OR to track plantain development throughout the butterfly lifecycle and the temporal pattern of *P. plantaginis* disease development. Two of the sites were locations of Taylor’s checkerspot populations and the other was a potential reintroduction site communicated by local land managers near Philomath, Oregon. We used random stratified sampling to locate monitoring plots at each of the three study sites. Within each site, we identified the area containing plantain and where oviposition is known to occur or likely to occur given the specific suite of conditions associated with oviposition [12] and divided the area into five, equally sized strata. Within each stratum, we randomly selected 3 locations from a 2 m × 2 m grid to provide dispersion across the site and equal probability of monitoring plot site selection. Plot locations were recorded with a handheld, WASS-enabled, Garmin eTrex 100 GPS unit, which consistently yields submeter accuracy [31] and the plot centers were marked with a 2-cm diameter plastic disc to aid in accurate plot relocation. Because we wanted to understand phenological development spanning ~450 km but could not accurately estimate disease severity in both Washington and Oregon without introducing significant levels of observer bias, we counted the number of plants with *P. plantaginis* infections (those characterized by foliar necrosis and the presence of apothecia) and those without infections to estimate the percentage of plants infected. However, *P. plantaginis* disease is at its apex in January, so what we effectively measured was how the diseased plantain population diminishes over time, beginning on 1-January. Using the percentage of plants with disease symptoms, which could be communicated by land managers/surveyors in Washington, we developed a coarse representation of how the disease presence diminishes with time over the butterfly lifecycle. Data from these monitoring plots were averaged over all years to represent disease overlap with butterfly development. To index *P. plantaginis* disease severity, we applied a cutoff point of 50% or greater leaf crown necrosis to represent a severe infection and tracked the reduction in plants with severe disease levels over time. The cutoff value of ≥50% leaf crown necrosis has biological significance because this was a level of disease that was mostly unacceptable for post-diapause larvae in experimental feeding trials [11]. By adopting this disease severity cutoff, we admittedly do not accurately represent the range of *P. plantaginis* disease severities but it is a relatively straightforward and quick measurement to take that is linked to important feeding larval behaviors.

In the same *P. plantaginis* monitoring plots, we also tracked the phenology of plantain leaf initiate production. Although Taylor’s checkerspot post-diapause larvae can feed on the previous growing year’s basal plantain leaves, larvae have a very strong feeding preference for leaf initiates [11]. Since plantain leaf initiates are highly valued by post-diapause larvae but can be delayed when *P. plantaginis* disease levels exceed 50% cover [11], leaf initiates were the resource we elected to track over time. To phenologically describe food plant resource availability, we recorded the number of plantain individuals with leaf initiates over time (~every 2–3 weeks) and presented them as the percentage of the total number of plants with new leaves from January through May and averaged values over the survey years. 

Although *Castilleja levisecta* is not known to occur naturally in Oregon Taylor’s checkerspot sites, two western Oregon restoration plantings that offered similar habitat conditions and elevation range to extant Taylor’s checkerspot populations were visited once every two weeks beginning with post-diapause larval activity in the winter of 2015 and 2016. Tracking was focused on determining when basal leaves emerged from the below-ground meristems (*C. levisecta* is a perennial herbaceous plant with deciduous leaves). Because *C. levisecta* were planted on grid points, individual plants were easily located and could be unambiguously tracked over time. Biologists in Washington communicated the approximate developmental phenology in their *C. levisecta* populations over the same time period.

*Collinsia parviflora* is an annual plant that typically emerges from seeds either germinating in the fall (shortly following the onset of autumn rains) or winter. In western Oregon grasslands, *C. parviflora* is rare and has never been recorded in Oregon Taylor’s checkerspot locations, presumably due to competition from invasive exotic grasses. In Washington, *C. parviflora* is relatively uncommon, except where bare soil is abundant (e.g., stabilized sand dunes and glacial outwash) [15]. To characterize *C. parviflora* phenology over the post-diapause larval feeding time, seeds from western Oregon and Washington were tracked in a common garden (private owned land in Philomath, Oregon). Two hundred seeds obtained from the only known low elevation *C. parviflora* site in western Oregon (the base of a rocky cliff in Linn County, OR) and 200 seeds from a western Washington glacial outwash location (vicinity of Mima Mounds, Thurston County, WA, USA) were evenly broadcast in the first week of July (2014) over two separate 1 m × 1 m common garden plots (a mixture of 80% potting soil and 20% sand) having similar aspect (% 5 slope and southwestern exposure) as the western Oregon butterfly populations. Plots were lined with sodium ferric EDTA (Corry’s Slug and Snail Killer™) in a 7-cm swath around the perimeter of the plot with a 10-cm buffer between the seed planting area and the inside edge of the molluscicide. Molluscicide was applied weekly (or as needed to maintain the 7-cm barrier) and plots were visited weekly beginning in early September, when there was no rain and temperatures >25 °C, through May of the next year. Naturally cycling *C. parviflora* seeds through the hot, dry summer (July through September) followed by autumn/winter rains (October–March), mimicked natural germination cues for both Oregon and Washington populations. A toothpick was used to mark the location of emerged seedlings to aid in counting as the number of *C. parviflora* germinants accumulated over time.

## 3. Results

Profiles of Taylor’s checkerspot butterfly development and foodplant availability indicated that there were important similarities and differences between Oregon (Figure 2) and Washington (Figure 3) populations. First, only plantain (in the absence of disease) provided sufficient food resources for Taylor’s checkerspot throughout its entire life cycle (Figure 2 and Figure 3). In contrast, no native foodplant alone provided food resources that overlapped all Taylor’s checkerspot developmental stages. Second, butterfly and foodplant development were approximately two to three weeks earlier in Oregon than in Washington (compare Figure 2 and Figure 3), which is not unexpected given that there is ~6 degrees of latitude (~450 km) separating these populations (Figure 1). Third, *Castilleja levisecta*, a focus for future Taylor’s checkerspot reintroductions by Oregon and Washington land managers, had scant aboveground tissues available for Taylor’s checkerspot post-diapause larvae to consume until late February or early March. Although there were small (<1 cm^2^) *C. levisecta* single leaves available at the base of some plants from January through mid-March (Oregon), the amount of biomass is about one hundredth of the single leaf area of *C. levisecta* in April/May (Severns pers. obs.). It is possible that these small leaves were initiated in the fall and not new leaf growth; their morphology did differ from the leaves emerging later in the growing season. However, *en masse*, *C. levisecta* leaf initiation did not begin until mid to late March (Figure 2 and Figure 3). Last, the germination profiles of *Collinsia parviflora* appeared to substantially differ between Oregon and Washington populations. Seeds from Oregon populations delayed germination until late February to early March (Figure 2), while the Washington *C. parviflora* had almost half of the seeds germinate in the fall, prior to post-diapause larval activity, with the remainder germinating in the late winter (Figure 3).

## 4. Discussion

Plantain’s occurrence as a predictable, perennially-abundant weed of disturbed and open habitats throughout North America [6] likely explains the original acquisition of plantain (prior to *P. plantaginis* invasion) by Taylor’s checkerspot butterfly over a century ago. The patterns of phenological development also suggest that the ancestral foodplants, prior to plantain acquisition, likely involved at least one of the *Castilleja* spp. and *Collinsia parviflora*. *Castilleja* spp. very likely served as the oviposition and pre-diapause larval foodplants, and this interaction has been recorded for Taylor’s checkerspot populations in Washington [15]. *Collinsia parviflora* appears necessary to provide enough food for post-diapause larvae until *Castilleja* produces spring leaves (Figure 2 and Figure 3). Presently, these native host plant interactions may only naturally occur in Washington as seeds from the Oregon population of *C. parviflora* did not begin to germinate until the late winter (compare Figure 2 and Figure 3), six weeks after post-diapause larvae first began actively feeding (Julian day 7 = mean date for first post-diapause larval feeding; range Julian day 359–18 in Oregon). 

As a sole host plant, *C. levisecta* appeared highly unlikely to support a Taylor’s checkerspot population due to at least six weeks of little or no food resources for post-diapause larvae. The same is likely true for *C. hispida*, another known Taylor’s checkerspot oviposition and larval foodplant [15]. *Castilleja hispida* and *C. levisecta* readily hybridize where they co-occur [32,33]. indicating that they have overlapping flowering times and likely similar life histories. 

Following a century or more of apparent dependence on plantain as the sole larval foodplant, most Taylor’s checkerspot populations face a substantial risk of extinction from *P. plantaginis*. It would only take a single year of or near 100% disease levels to result in butterfly population extinction, and it appears this may have already happened to a naturally occurring population and a butterfly reintroduction project [11]. It is common for plant pathogens and their associated diseases, to fluctuate in relative abundance and disease severity between years due to specific suites of environmental conditions that can encourage or depress pathogen levels, e.g., [34,35]. The expectations for *P. plantaginis* should be no different and it is the sudden and severe disease increases that are the threat to Taylor’s checkerspot. Although, cool temperatures above freezing and high relative humidity appeat to favor infection and disease, which is consistent with reports from Europe [8], the precise conditions for *P. plantaginis* infection, disease intensification, and dispersal remain mostly unknown. 

For Taylor’s checkerspot butterfly, the solution to *P. plantaginis* disease appears to be conceptually simple—supplement habitat with native foodplant species that are options for post-diapause larvae and apply fungicides to reduce *P. plantaginis* disease. However, there are some practical and legal challenges to these potentially straightforward management interventions. First, because Taylor’s checkerspot is a federally listed endangered species, it will be difficult to manage *P. plantaginis* through fungicides, as one would in agricultural systems, because there may be unintended fungicide effects that either directly harm the butterfly (considered “a take” under the Endangered Species Act) or the native plant community the butterfly depends upon [36,37]. One such possibility is that fungicides could interfere with the obligate mycorrhizal associations that enable *Castilleja* spp. to be a hemi-parasite on their host plant(s). Second, to feature any *Castilleja* spp. as the dominant larval foodplant, it is clear that without either *Collinsia parviflora* or plantain, post-diapause larvae would be highly unlikely to survive from January through February (Figure 2 and Figure 3). Supplementation of habitat with *Collinsia parviflora* presents a considerable challenge because it is an annual that requires bare ground with low plant competition to generate enough biomass to support Taylor’s checkerspot. Only one extant Taylor’s checkerspot population is known where this condition is fulfilled and it is a stabilized sand dune near Sequim, WA where *C. parviflora* forms dense carpets of vegetation [15]. *C. parviflora* protects this butterfly population from *P. plantaginis* as disease levels were relatively high at this site [11] but the butterfly population flourishes [15]. All extant butterfly sites in western Oregon are not suitable for *C. parviflora* as bare ground is rare, plant competition is high, and the dominant, exotic grasses are themselves a direct threat to butterfly persistence [12]. Furthermore, the *C. parviflora* populations in Oregon may not be appropriately timed for Taylor’s checkerspot post-diapause larvae (Figure 2). If this is true, then habitat restoration in Oregon would require *C. parviflora* from Washington (where germination is appropriately timed). However, such long-distance transfer of *C. parviflora* genetic resources may not be philosophically or biologically acceptable to land managers. Land managers should consider researching the safe application of fungicides and establishing butterfly-sustaining populations of alternative host plants to manage *P. plantaginis* disease and mitigate for severe disease outbreaks, regardless of whether such actions may be counter to typical conservation paradigms. 

## 5. Conclusions

Although it seems that the initial acquisition of plantain by Taylor’s checkerspot was likely an adaptive switch to a more dependable and predictable food resource, the recent invasion of *P. plantaginis* now poses considerable risk to the long-term persistence of plantain-dependent butterfly populations. Our phenological profiles, while not intended to be exact and represent all past and future weather/climate situations, do indicate the challenges posed for Taylor’s checkerspot butterfly populations from mismatches in butterfly development, host plant availability, and the presence of plant disease. *Pyrenopeziza plantaginis* invasion has modified the co-evolved interactions between Taylor’s checkerspot and plantain and consequently re-elevated the importance of the ancestral native larval foodplants from which the butterfly evolved. It is unclear whether *P. plantaginis* will eventually lead to the extinction of plantain-dependent Taylor’s checkerspot populations, or what specific environmental conditions would promote such an event. However, understanding the basic epidemiological factors influencing *P. plantaginis* disease infection, intensification, and spread will be crucial for future butterfly conservation practices and its long-term persistence.

## Figures and Tables

**Figure 1 insects-12-00246-f001:**
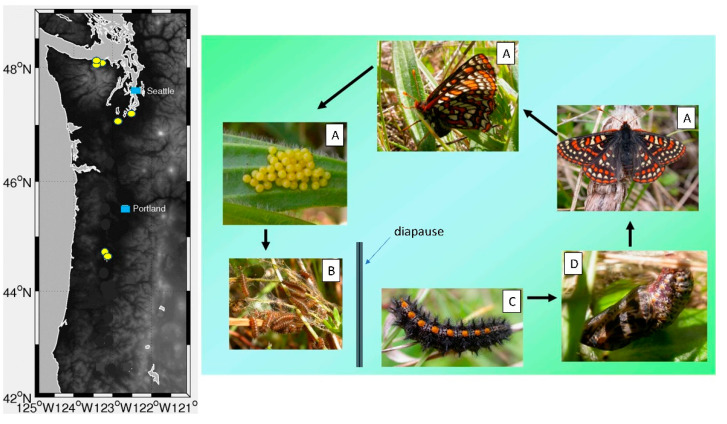
Left: Map of extant Taylor’s checkerspot populations (yellow circles) in Washington and Oregon, USA. Major cities Seattle, Washington and Portland, Oregon are represented by blue squares. Right: Life cycle of Taylor’s checkerspot butterfly and general development timing. Life stages (**A**) appearance of reproductive adult butterflies and oviposition on the pre-diapause larval foodplant (April-May). (**B**) Pre-diapause larvae feed on leaves within webbed shelters spun on the oviposition plant (May-June) and enter diapause in July. (**C**) Post-diapause larvae resume feeding activity in the winter (January-March) and (**D**) pupation occurs in March-April.

**Figure 2 insects-12-00246-f002:**
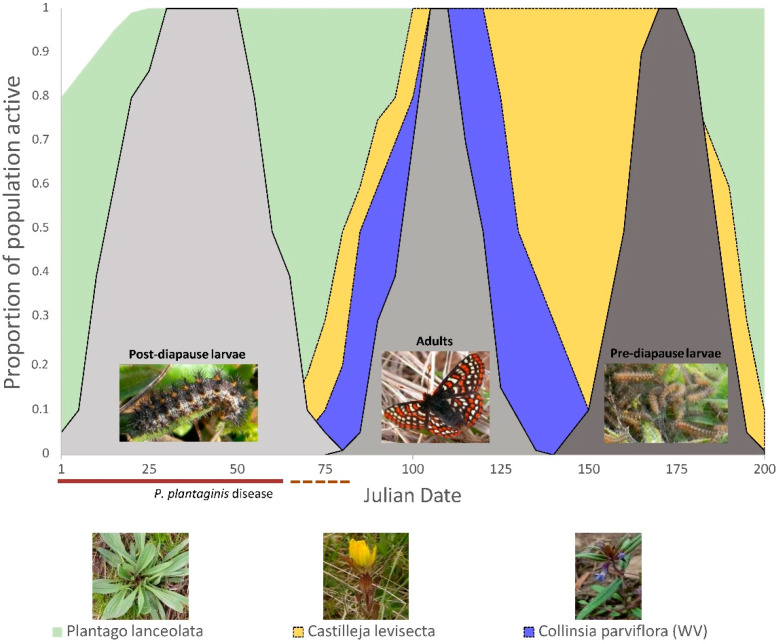
Temporal profiles of plantain-dependent Taylor’s checkerspot butterfly development and host plant availability arranged by Julian date in Oregon, USA populations. Relative abundance of butterflies (gray shaded areas) and host plants (green = *Plantago lanceolate*—plantain, blue = *Collinsia parviflora*, yellow = *Castilleja levisecta*) are represented as the proportion of the total observed for that life stage (butterflies) or monitored plant populations. *P. plantaginis* disease occurrence is indicated by the bar below the *x*-axis where the time period with the most severe disease (>50% of diseased plants with >50% foliar necrosis) is represented by a solid brown line and a dotted brown line represents the transition period from severe disease (>50% necrosis covering the entire leaf crown of infected plants) to no disease.

**Figure 3 insects-12-00246-f003:**
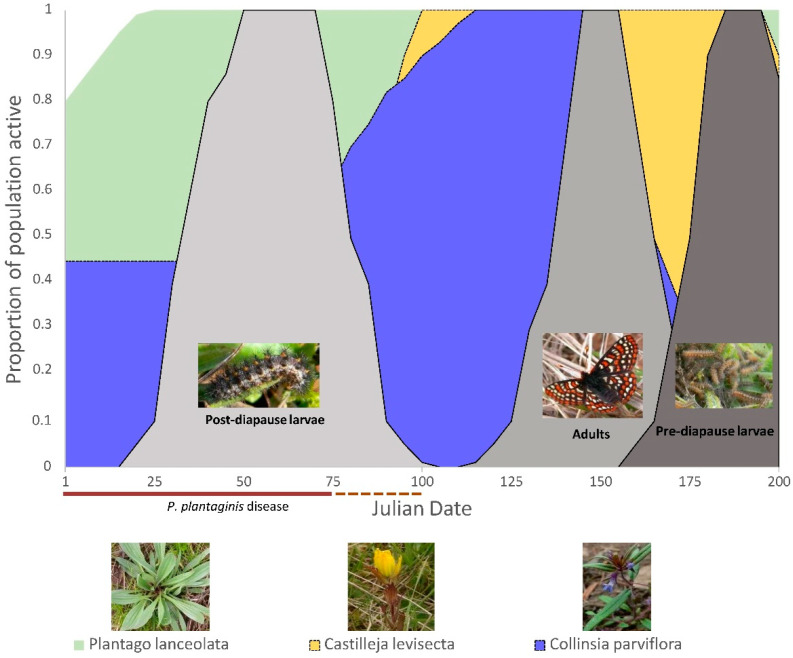
Temporal profiles of plantain-dependent Taylor’s checkerspot butterfly development and host plant availability arranged by Julian date in Washington, USA populations. Relative abundance of butterflies (gray shaded areas) and host plants (green = *Plantago lanceolate*—plantain, blue = *Collinsia parviflora*, yellow = *Castilleja levisecta*) are represented as the proportion of the total observed for that life stage (butterflies) or monitored plant populations. *P. plantaginis* disease occurrence is indicated by the bar below the *x*-axis where the time period with the most severe disease (>50% of diseased plants with >50% foliar necrosis) is represented by a solid brown line and a dotted brown line represents the transition period from severe disease (>50% necrosis covering the entire leaf crown of infected plants) to no disease.

## Data Availability

Data are available upon request from the authors.

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
