# Peer review of "Plant Pathogen Invasion Modifies the Eco-Evolutionary Host Plant Interactions of an Endangered Checkerspot Butterfly"

_insects, 2021, doi:10.3390/insects12030246_

Round 1
Reviewer 1 Report
The authors provide a very sound and well-presented manuscript regarding a relatively new pathogen impacting non-native Plantago that directly impacts extant populations of the federally endangered Taylor's checkerspot butterfly- and present implications to its long-term viability in Washington and Oregon. As such, this is very important research and the findings have direct implications to conservation, management and restoration efforts/planning for this imperiled taxon. Overall, I have very few suggestions or comments. Please see them on the manuscript itself. Outside of a few minor grammatical edits and sentence re-writes for clarity, I suggest that more detail be presented in the Methods regarding monitoring plot establishment. I also suggest that the authors take the opportunity to directly suggest the need for additional research to address potential management for this butterfly (i.e. fungicide trails, restoration plots with alternative hosts regarding plant density and resources, etc.). I also question the organization of the Methods section a bit, particularly the first life history component.

Author Response
Reviewer #1
The authors provide a very sound and well-presented manuscript regarding a relatively new pathogen impacting non-native Plantago that directly impacts extant populations of the federally endangered Taylor's checkerspot butterfly- and present implications to its long-term viability in Washington and Oregon. As such, this is very important research and the findings have direct implications to conservation, management and restoration efforts/planning for this imperiled taxon. Overall, I have very few suggestions or comments. Please see them on the manuscript itself. Outside of a few minor grammatical edits and sentence re-writes for clarity, I suggest that more detail be presented in the Methods regarding monitoring plot establishment. I also suggest that the authors take the opportunity to directly suggest the need for additional research to address potential management for this butterfly (i.e. fungicide trails, restoration plots with alternative hosts regarding plant density and resources, etc.). I also question the organization of the Methods section a bit, particularly the first life history component.
Lines 63-65 suggest that the authors avoid repetition of the organism name twice in a sentence and instead use al alternative such as taxon
We have edited this sentence as suggested
Line 89 Change “of” to “on”.
This has been changed.
Line 91: add “on” between “feed partially”
This has been changed.
Lines 112-125: While this information is important, I question if it should go in the Materials and Methods section here. This is background information but seems a bit out of place under this heading.
Thank you for calling this to our attention. We have reworded the subheading to be more inclusive of the topics covered. The subheading now reads, “Taylor’s checkerspot geographic distribution, developmental timing, and life history”
Line 120: This seems like an odd reference. Can just say "on the plant"
We have modified this sentence to indicate that the pre-diapause larval masses can move to and feed on plants other than the oviposition plant. The sentence now reads, “Eggs develop over two to three weeks and then hatch (May-June), giving rise to clusters of 1st instar larvae that feed gregariously (initially on the oviposition plant and later on nearby plants) through three instars (about 1 month) and then enter diapause for the remainder of the summer and fall (Figure 1).”
Lines 124-125: This sentence is somewhat confusing, suggest rewriting for clarity.
Thank you for bringing this confusing passage to our attention. We combined two sentences to make the following passage which now ends the paragraph in a more concise and clear manner. “Post-diapause larvae become active and begin feeding in the winter, although the earliest dates of larval feeding are generally not reported as most butterfly field work begins in March when larvae are in their last (5th) instar.”
Line 174: Insert “larval feeding” between post-diapause and begins.
We have made this change.
Line 174: Replace begins with began
Done
Lines 191-193: It is important to describe how plot locations were determined so the study could be replicated.
Thank you for bringing to our attention that we did not adequately describe the spatial plot location selection method. We have added three sentences describing the plot location method. The new passage follows: “We used random stratified sampling to locate monitoring plots at each of the three study sites. Within each site, we identified the area containing plantain and where oviposition is known to occur or likely to occur given the specific suite of conditions associated with oviposition [12] and divided the area into five, equally sized strata. Within each stratum, we randomly selected 3 locations from a 2m x 2m grid to provide dispersion across the site and equal probability of monitoring plot site selection.”
Lines 289-291: This sentence is also a bit unclear, suggest revising.
We have rephrased the first sentence of the Introduction to the following to generate a more clearly stated passage: “Plantain’s occurrence as a predictable, perennially-abundant weed of disturbed and open habitats throughout North America [6] likely explains the original acquisition of plantain (prior to P. plantaginis invasion) by Taylor’s checkerspot butterfly over a century ago.”
Line 322: suggest saying federally listed instead of protected
Done
Lines 322-342: These are ideal suggestions for next research steps that could be experimentally addressed using field plots and possibly surrogate butterfly taxa. I would suggest that the authors specifically suggest the need for future research and list these key data gaps.
We have added a sentence prior to the section 5 (conclusions) to encourage research into the use of fungicides and alternative host plants even though this approach may be counter to the traditional conservation biology paradigms. The new sentence reads,” Land managers should consider researching the safe application of fungicides and establishing butterfly-sustaining populations of alternative host plants to manage P. plantaginis disease and mitigate for severe disease outbreaks, regardless of whether such actions may be counter to typical conservation paradigms.”
Reviewer 2 Report
The paper is excellent but needs a little further proofreading, In several instances, words are deleted from sentences or an improper construction is used. Here are some examples I noted:
are highly unlikely to feed partially symptomatic leaves, including the green portions that appear 91
One more careful proofing and correction should correct these minor problems.
Washington the same as in Oregon, we simply recorded the number of plants with P. plantaginis 195
acquisition of plantain (prior to P. plantaginis invasion) by Taylor’s checkerspot butterfly as it a 290
Our phenological profiles, while not intended to be exact and represent all past 347 and future weather/climate situations, they do indicate the challenges posed for Taylor’s checkerspot butterfly populations. 348/349
lations or what conditions such an event might occur. However, understanding P. plantaginis epi- 353
One more careful proofing and correction should take care of these minor problems.
Author Response
Reviewer #2
The paper is excellent but needs a little further proofreading, In several instances, words are deleted from sentences or an improper construction is used. Here are some examples I noted:
are highly unlikely to feed partially symptomatic leaves, including the green portions that appear 91
This sentence was rewritten and is reported in the response to reviewer #1.
One more careful proofing and correction should correct these minor problems.
We have proofread the manuscript paying attention to minor grammatical issues and sentence syntax. Those changes are small in number so we do not detail the changes as they are minor issues but they are represented in the marked up version of the manuscript.
Washington the same as in Oregon, we simply recorded the number of plants with P. plantaginis 195
We added information to add clarity to this sentence. The passage now reads: “Because we wanted to understand phenological development spanning ~ 450 km but could not accurately estimate disease severity in both Washington and Oregon without introducing significant levels of observer bias, we counted the number of plants with P. plantaginis infections (those characterized by foliar necrosis and the presence of apothecia) and those without infections to estimate the percentage of plants infected.”
acquisition of plantain (prior to P. plantaginis invasion) by Taylor’s checkerspot butterfly as it a 290
We revised this sentence, and the revised sentence is provided in our response to reviewer #1.
Our phenological profiles, while not intended to be exact and represent all past 347 and future weather/climate situations, they do indicate the challenges posed for Taylor’s checkerspot butterfly populations. 348/349
We have revised this sentence which now reads: “Our phenological profiles, while not intended to be exact and represent all past and future weather/climate situations, do indicate the challenges posed for Taylor’s checkerspot butterfly populations from mismatches in butterfly development, host plant availability, and the presence of plant disease.” We hope this revision provides the degree of clarity reviewer 2 was seeking.
lations or what conditions such an event might occur. However, understanding P. plantaginis epi- 353
We have modified the final two sentences of the manuscript to the following: “It is unclear whether P. plantaginis will eventually lead to the extinction of plantain-dependent Taylor’s checkerspot populations, or what specific environmental conditions would promote such an event. However, understanding the basic epidemiological factors influencing P. plantaginis disease infection, intensification, and spread will be crucial for future butterfly conservation practices and its long-term persistence.” We hope this improves the passage in the spirit of reviewer 2’s comment.
Reviewer 3 Report
This paper addresses an intriguing situation in which an endangered butterfly species has become dependent on an invasive plant that is itself now threatened by an invasive pathogen. This may be of broader interest in the conservation community as yet another example where the management of endangered and invasive species are at odds. I only have two concerns.
First, there are little errors throughout, extra commas, extra periods such as in Figure 2 legend, wrong words (91 "feed" should be "eat the") and occasional awkward and run-on sentences (e.g., starting on 289, 314, 327). The pre-diapause butterflies in Figs 1 and 2 are not gray, as stated in the legends and the populations described are not identified. The summary gives Latin names for two host plants but only common names for the butterfly and invasive host plant, and the label "pathogen". The abstract corrects some of these problems and identifies the pathogen by Latin name but not as a ascomycete. This manuscript would benefit from a careful friendly editing.
Second, some important details are omitted. There apparently are eight butterfly populations remaining in Oregon and Washington and the authors focused on the lowland ones, but how many lowland populations were there and which were studied? Some populations were monitored from 2008 through 2017 though the main Figures 2 and 3 appear to be averages or composites for just 2014-2016. Is that correct? If there are several years of data summarized in the figures, why not present the results for each year separately? It would be extremely helpful to know how much annual variation there is in the phenology of the butterfly and fungal infestations of their foodplants, and the extent to which such variations are in lockstep.
I very much appreciate the comments at the end on possible solutions and wonder, if the populations are localized enough, whether the removal of infected leaves after the butterflies have finished feeding would be another possible solution.
Author Response
Reviewer # 3
This paper addresses an intriguing situation in which an endangered butterfly species has become dependent on an invasive plant that is itself now threatened by an invasive pathogen. This may be of broader interest in the conservation community as yet another example where the management of endangered and invasive species are at odds. I only have two concerns.
First, there are little errors throughout, extra commas, extra periods such as in Figure 2 legend, wrong words (91 "feed" should be "eat the") and occasional awkward and run-on sentences (e.g., starting on 289, 314, 327).
We have revised the passage beginning at line 289 and detailed the changes in the response to Reviewer #1.
We have edited the passage beginning at line 314 and broken the passage into two smaller sentences.
We have edited the sentence beginning at line 327 and shortened it to the following: “Although, cool temperatures above freezing and high relative humidity favor infection and disease, which is consistent with reports from Europe [8], the conditions for P. plantaginis infection, disease intensification, and dispersal remain mostly unknown.”
We have modified the passage in question on Figure 1 to now read: “Pre-diapause larvae feed on leaves within webbed shelters spun on the oviposition plant (May-June) and enter diapause in July.”
The pre-diapause butterflies in Figs 1 and 2 are not gray, as stated in the legends and the populations described are not identified.
There may be an issue with reviewer 3’s screen or ink if the manuscript was printed, because the different butterfly developmental profiles are definitely gray on all downloaded copies and Figures 2 and 3 (the only figure with gray, not Figure 1 as the reviewer indicated), were made by selecting colors from the gray shade panel.
The summary gives Latin names for two host plants but only common names for the butterfly and invasive host plant, and the label "pathogen". The abstract corrects some of these problems and identifies the pathogen by Latin name but not as a ascomycete. This manuscript would benefit from a careful friendly editing.
We added “(Dermateaceae)” to the abstract to indicate P. plantaginis’ higher level taxonomic placement.
Second, some important details are omitted. There apparently are eight butterfly populations remaining in Oregon and Washington and the authors focused on the lowland ones, but how many lowland populations were there and which were studied? Some populations were monitored from 2008 through 2017 though the main Figures 2 and 3 appear to be averages or composites for just 2014-2016. Is that correct? If there are several years of data summarized in the figures, why not present the results for each year separately? It would be extremely helpful to know how much annual variation there is in the phenology of the butterfly and fungal infestations of their foodplants, and the extent to which such variations are in lockstep.
The site names for the lowland populations were named in the last paragraph of the Introduction section and now included in the methods section.
We wish we had more precise timing data, the monitoring at all sites is unpredictable as weather and site access issues in some populations allow for only 3 visits in any given year. We have indicated this as an issue in the methods section in the first paragraph of subsection 2.2. The new passage reads as follows: “Adult Taylor’s checkerspot surveys in both states can be acutely limited by access restrictions and inclement weather, such that some sites receive as little as 3 visits in a flight season, yet others will have 7 surveys. This data set presents some challenges as it can be sparse depending upon site and year, so we focus on the broad phenological development signals.”
We understand the concerns of precise phenological profile representation. Unfortunately, our data are not resolved enough to be confident about anything but the average patterns. Initial attempts to represent the variation in the phenological profiles were so busy as to be nearly unreadable, even for the authors. This is in part due to the patchwork nature of the combined data set. We did the best we could and agree that more precision would be ideal. We are straightforward in the conclusions section that we are aware that our phenological profiles do not capture environmental variation.
I very much appreciate the comments at the end on possible solutions and wonder, if the populations are localized enough, whether the removal of infected leaves after the butterflies have finished feeding would be another possible solution.
Reviewer 4 Report
Because responsiveness to iridoid glycosides is fairly generalized, but especially the work of Singer's group shows that local Euphydryas populations may specialize and the growth form pf the plant is a significant recognition sign, a search should be mounted for acceptable alternate hosts that could be naturalized in the affected areas and will sustain larval development during the critical period. They can then be field-tested. Are other plantagos susceptible to the pathogen? P.lanceolata seems to have a heavier dose of iridoids than the broad-leafed weeds, major and rugelii, but they can be checked out first.
A few editorial items: line 11, over a century' l.39, suitably timed' l.41, delete "to"; l.58, delete comma;l.91, feed on; l.213, elected; l.353, under what conditions.
Author Response
Because responsiveness to iridoid glycosides is fairly generalized, but especially the work of Singer's group shows that local Euphydryas populations may specialize and the growth form pf the plant is a significant recognition sign, a search should be mounted for acceptable alternate hosts that could be naturalized in the affected areas and will sustain larval development during the critical period. They can then be field-tested. Are other plantagos susceptible to the pathogen? P.lanceolata seems to have a heavier dose of iridoids than the broad-leafed weeds, major and rugelii, but they can be checked out first.
We honestly do not understand how reviewer 4 would like us to integrate these comments into the manuscript, nor necessarily understand why the iridoid glycosides are the focus of the comments. We did not measure iridoid glycosides so we are confused by the reviewer’s comments.
We can address the question about P. plantaginis infecting other Plantago species. It is not clear whether other Plantago species are infected by P. plantaginis. Possibly, but P. plantaginis is not a fungal disease on an economically valuable plant nor on plants that are ecologically important to ecosystem functioning, so very little is known about this particular pathogen. We refer the reader to reference [8] frequently throughout the manuscript, which summarizes what is known about P. plantaginis. Which is to say, very little. We are non-committal on this account because we can’t be sure about either P. plantaginis being restricted to P. lanceolata or if it broadly occurs on other Plantago spp. and causes severe disease as it does in P. lanceolata.
A few editorial items:
line 11, over a century'
done
l.39, suitably timed'
done
l.41, delete "to";
done
l.58, delete comma;
done
l.91, feed on;
done
l.213, elected;
done
l.353, under what conditions.
We revised this passage and detail the changes in response to reviewer #2’s comments.